# Early heart rate variability changes during acute fetal inflammatory response syndrome: An experimental study in a fetal sheep model

**Geoffroy Chevalier** [1,2]*, **Charles Garabedian**[1,2], **Jean David Pekar**[3], **Anne Wojtanowski**[4], **Delphine Le Hesran**[2], **Louis Edouard Galan**[2], **Dyuti Sharma**[1,5], **Laurent Storme**[1,6], **Veronique Houfflin-Debarge**[1,2], **Julien De Jonckheere**[1,4], **Louise Ghesquière** [1,2]

**1** ULR 2694—METRICS—Evaluation des Technologies de Santé et des Pratiques Médicales, University Lille, CHU Lille, France, **2** Department of Obstetrics, CHU Lille, France, **3** Automated Biochemistry (UF 8832), CHU Lille, France, **4** CIC-IT 1403, CHU Lille, France, **5** Department of Pediatric Surgery, CHU Lille, France, **6** Department of Neonatology, CHU Lille, France

* geoffroy.chevalier@chu-lille.fr

**Data Availability Statement:** All relevant data are within the manuscript and supporting information.

## Abstract

### Introduction

Fetal infection during labor with fetal inflammatory response syndrome (FIRS) is associated with neurodevelopmental disabilities, cerebral palsy, neonatal sepsis, and mortality. Current methods to diagnose FIRS are inadequate. Thus, the study aim was to explore whether fetal heart rate variability (HRV) analysis can be used to detect FIRS.

### Material and methods

In chronically instrumented near-term fetal sheep, lipopolysaccharide (LPS) was injected intravenously to model FIRS. A control group received saline solution injection. Hemodynamic, blood gas analysis, interleukin-6 (IL-6), and 14 HRV indices were recorded for 6 h. In both groups, comparisons were made between the stability phase and the 6 h following injection (H1–H6, respectively) and between LPS and control groups.

### Results

Fifteen lambs were instrumented. In the LPS group (n = 8), IL-6 increased significantly after LPS injection (p < 0.001), confirming the FIRS model. Fetal heart rate increased significantly after H5 (p < 0.01). In our FIRS model without shock or cardiovascular decompensation, five HRV measures changed significantly after H2 until H4 in comparison to baseline. Moreover, significant differences between LPS and control groups were observed in HRV measures between H2 and H4. These changes appear to be mediated by an increase of global variability and a loss of signal complexity.

### Conclusion

As significant HRV changes were detected before FHR increase, these indices may be valuable for early detection of acute FIRS.

Compete data about each fetus is provided in Supporting information.

**Funding:** The author(s) received no specific funding for this work.

**Competing interests:** The authors have declared that no competing interests exist.

## Introduction

Intrauterine infection and inflammation (III) are known as risk factors for neonatal brain damage, morbidity, and mortality [1]. III can be associated with a fetal inflammatory response syndrome (FIRS), a systemic inflammation in which fetal plasma interleukin-6 (IL-6) is elevated [2]. During FIRS, increased IL-6 and other proinflammatory cytokines have been implicated in the development of periventricular leukomalacia, cerebral palsy, neonatal sepsis, and mortality [1,3–7]. A large cohort study showed that during labor, maternal fever (an objective sign of III) without fetal hypoxia was associated with neonatal encephalopathy (odds ratio [OR] = 6.3 [2.7–14.8]), and maternal fever associated with fetal hypoxia was strongly associated with neonatal encephalopathy (OR = 76.2 [23.1–251.7]) [8].

However, clinical signs of III show poor or limited sensitivity and specificity [5]. Despite the morbidity and mortality associated with FIRS, current methods to detect fetal infection are inadequate [5]. Improved detection of the early signs of FIRS would be valuable for identifying fetal complication risks. If identified, these fetuses at risk of morbidity could benefit from more vigilant monitoring during labor. A suspicion of concomitant hypoxia could lead to early cesarian or instrumental delivery.

Heart rate variability (HRV) could be an interested tool to detect FIRS. HRV reflects changes in time intervals between consecutive heartbeats [9]. It is an efficient way to study the autonomic nervous system (ANS) which regulates systemic infection and inflammation [10]. Activation of the adrenergic system during sepsis is critical for initiating a physiologic response to pathogens but can become detrimental in excess. In contrast, the efferent vagus nerve inhibits proinflammatory cytokine release and protects against systemic inflammation. This vagal function is called the cholinergic anti-inflammatory pathway [11].

It was previously shown that detection of neonatal infection is possible using abnormal heart rate characteristics including HRV[12]. Perinatal studies show that HRV monitoring is a potential noninvasive, sensitive, and specific measure of inflammatory response [13,14]. In a randomized controlled trial of neonates in intensive care, HRV monitoring was associated with lower septicemia, possibly due to earlier diagnosis of illness [14].

Since HRV is correlated with SNA alterations and since it allows detection of neonatal inflammation, we hypothesized that HRV analysis can be used for early detection of acute FIRS. The study aims to explore HRV indices response to acute FIRS in a fetal sheep model.

## Material and methods

### Surgical preparation

Near-term pregnant sheep (race 'Ile de France', Tours, INRA, Orfrasière Animal Physiology Experimental Unit, Val de Loire Center) of gestational age 124 ± 1 d (term = 145 d) underwent our previously described surgical procedure [10,15,16]. Briefly, sheep were fasted for 24 h before general anesthesia and surgery. They were then placed supine, anesthetized with an intravenous injection of xylazine (Sedaxylan®; CEVA Santé Animale, Libourne, France), intubated, and maintained with 2% isoflurane (Aerrane®; Baxter, Guyancourt, France). After maternal laparotomy and hysterotomy, catheters (umbilical catheters 4Fr diameter, Vygon, France, Ecouen) were placed in the fetal left axillary artery and vein and in the right axillary artery. Four electrocardiogram electrodes (Mywire 101; Maquet, Rastatt, Germany) were placed on the fetal intercostal muscles near the heart to record fetal electrocardiogram. A 5F5-diameter catheter (Arrow®) was placed into the amniotic cavity to replace amniotic fluid lost during surgery with 500 mL saline containing antibiotics (amoxicillin–clavulanic acid) and to measure baseline intra-amniotic pressure. All leads were exteriorized through the

maternal flank. After surgery, ewes were given free access to food and drink. Postoperative analgesia was provided by maternal intramuscular injection of 0.3 mL/10 kg buprenorphine (Buprenodale®; Dechra Veterinary Products, Montigny-le-Bretonneux, France) at 24 and 48 h after surgery.

## Data acquisition

Fetal arterial and intra-amniotic catheters were connected to pressure sensors (Pressure Monitoring Kit®; Baxter). Blood pressure sensors and electrocardiogram electrodes were connected to a multiparametric anesthesia monitor (Merlin; Hewlett Packard, Palo Alto, CA, USA). Mean arterial pressure (MAP) was measured from blood pressure phasic signals and corrected for intra-amniotic pressure value (calculated MAP = observed MAP − observed intra-amniotic pressure). Electrocardiogram and blood pressure signals were recorded through a Physiotrace™ data acquisition board (Estaris Monitoring, Lille, France).

## Experimental procedure

The experiments began after the sheep had rested for at least 72 h after surgery. Lipopolysaccharide (LPS) derived from Escherichia coli, serotype 0111:B4 (Sigma-Aldrich, Merck, Darmstadt, Germany) was used to create a FIRS model.

Before LPS injection, a 60-min stability period was recorded to ensure that the animals were healthy (normal gas blood and normal hemodynamic parameters). Hemodynamic (MAP, fetal heart rate [FHR]), gasometric (pH, lactate, pO2, and pCO2), and HRV measures were recorded at the end of the stability period to obtain baseline values.

Eight fetuses received LPS (400 ng dissolved in 2 ml saline) intravenously to induce FIRS. This protocol was used by Durosier et al. to model FIRS in 10 fetal sheep without shock or cardiovascular decompensation [17]. Eight control fetuses received an equivalent volume of 0.9% NaCl solution.

FHR and arterial blood pressure were monitored continuously during the stability period and for 6 h after LPS or saline injection. Blood samples (2 mL) were collected for arterial blood gases, lactate, and IL-6 analyses at time points 0 (baseline) and 1 (H1), 2 (H2), 3 (H3), 4 (H4), 5 (H5), and 6 (H6) h after LPS or saline injection.

Euthanasia was administered at the end of the experimental procedure, or in case of fetal death. Euthanasia was carried out by intravenous injection of 6 ml/50 kg T61 (1 ml contains embutramide 200 mg + mebezonium 26.92 mg + tetracaine 4.39 mg, MSD, France).

## HRV analysis

ECG analysis to compute fetal R–R series was conducted offline using an automatic R-wave detection algorithm. HRV indices were computed through a program developed in MATLAB (version R2017B, MathWorks, Inc., Natick, MA, USA). Continuous computation of HRV indices uses a one-second moving window. Indices are then taken at average of the last 20 min.

HRV time domain analyses included: 1/ root mean square of successive differences (RMSSD) between adjacent R–R intervals; 2/ standard deviation of normal to normal R–R intervals (SDNN); 3/ short-term variability (STV), defined as the mean difference between successive 3.75-sec R–R interval epochs; and 4/ long-term variability (LTV), defined by the difference between the highest and lowest values within the 16 epochs of an analyzed minute [18].

Spectral HRV analysis included the low frequency (LF) component, from 0.04–0.15 Hz, which is related to both sympathetic and parasympathetic activity, and also associated with baroreflex activity. We also studied the high frequency (HF) component >0.15 Hz, which is

related to the parasympathetic nervous system alone. The LF/HF ratio represents parasympathetic–sympathetic imbalance [19].

Nonlinear analyses included: 1/ Poincaré plot standard deviation perpendicular to the line of identity (SD1); 2/ Poincaré plot standard deviation along the line of identity (SD2); 3/ approximate entropy (ApEn), which measures the regularity and complexity of a time series; 4/ detrended fluctuation analysis (DFA) $\alpha$1, which describes short-term fluctuations; and 5/ DFA $\alpha$2, which describes long-term fluctuations. These nonlinear measurements allow quantification of time series unpredictability [19]. A Poincaré plot is graphed by plotting every R–R interval against the prior interval, creating a scatter plot [19]. A Poincaré plot can be analyzed by fitting an ellipse to the plotted points. The standard deviation of each point from the $y = x$ + *average R–R interval* (SD2) specifies the ellipse's length [19] and SD2 measures short- and long-term HRV and is correlated with SDNN [20]. DFA is a nonlinear method for quantifying a fractal scale and the degree of correlation with an HRV signal in the form of a dimensionless measurement. Briefly, the root mean square fluctuation of the integrated and detrended data is measured in observation windows of different sizes. The data are then plotted against the size of the window on a log–log scale. The scaling exponent represents the slope of the line, which relates (log) fluctuation to (log) window size [21].

The last HRV indice was the Fetal stress index (FSI). It was developed by our team based on an original HRV analysis method that combines spectral and time domain analyses. Briefly, The RR series is isolated in a 64-second moving window, normalized and high pass-filtered above 0.15Hz using a wavelet-based numerical filter. The magnitudes of the remaining oscillations are computed by plotting local minima and maxima. The area between the upper and lower envelopes is divided into four sub-areas, A1, A2, A3 and A4, and the minimum area under the curve (AUC min) is defined as the minimum of the four sub-areas. A linear transformation is then applied to AUCmin to obtain an FSI value between 0 and 100: FSI = a × AUCmin + b, where a = 39.84 and b = 9.38 are two constants empirically determined on a dataset of 200 RR series records [22]. In previous experimental studies, we demonstrated that FSI was correlated with acidosis and parasympathetic activation [10,15, 16,23,24].

## Fetal arterial blood samples

Arterial blood gas parameters were measured with the i-STAT 1 blood analyzer (i-STAT 1 System; Abbott Point of Care, Inc., Princeton, NJ, USA) using CG4+ cartridges at specific time points.

Serum IL-6 concentrations were determined using an ovine-specific sandwich ELISA (ELISA KIT for interleukin 6, SEA079ov, Cloud-Clone Corp., Katy, TX, USA). Collected blood samples were centrifuged at 3500 g for 10 min and supernatant was stored at −80˚C until assessment. The kit microplate is pre-coated with an antibody specific to IL-6. Following the manufacturer instructions, biotin-conjugated antibody specific to IL-6 (100 μl) was added to the microplate wells and incubated for 1 h at 37˚C. After 3 washing stages, avidin conjugated to horseradish peroxidase was added to each microplate well and incubated for 30 min at 37˚C. After 5 washing stages, 5'-tetramethylbenzidine substrate solution (90 μl) was added and incubated for 10–20 min at 37˚C. The enzyme-substrate reaction was terminated by the addition of sulfuric acid solution (50 μl). Finally, optical density was measured spectrophotometrically at a wavelength of 450 nm.

The detection range of IL-6 was 7.8–500 pg/ml with an intra-assay precision of 10% and an inter-assay precision of 12%. When the maximum limit was reached, dilution was carried out with phosphate-buffered saline. Maximum dilution was 1/20.

## Statistical analyses

Numerical data are described as median (first and third quartiles). Differences between measures before and after LPS injection (H0 to H6) were evaluated using a Friedman nonparametric test for repeated measurements, followed by a Wilcoxon test when deemed significant (p < 0.05). Comparisons between LPS and control groups were performed by Mann-Whitney test. Statistical significance was assumed for p < 0.05. Data were analyzed using RStudio software (RStudio version 2022.07.2–576, USA).

## Ethics statement

Anesthesia, surgery, and experiment protocols were consistent with the recommendations of the French Ministry of Higher Education and Research, and the study was approved by the Animal Experimentation Ethics Committee (CEEA #2016121312148878).

# Results

## Cohort characteristics

Twenty-two fetuses were instrumented. Two fetuses died during the instrumentation, one died at the third post operative day. Nineteen experiments were performed: ten in LPS group and nine in the control group. In LPS group, one fetus died during the experimentation and one was excluded for severe intra uterine growth restriction. In the control group, one fetus was excluded for recording defect of HRV and a second for elevated Il-6 at baseline. In total, Eight LPS group and seven control group fetuses were included in analyses.

Median maternal body weight was 75.5 kg [72.5; 78.0]. Median fetal body weight was 3830 g [3625; 4092]. Gestational age at the experimental procedure was precisely 129 d for all fetuses. In the control group, 5/7 fetuses were male and 2/7 were singletons. In the LPS group, 6/8 fetuses were male and 3/8 were singletons.

Comparison of hemodynamic, blood gas and biochemical measures in LPS group from H1 to H6 compared with stability (H0), and comparison between LPS and control groups are shown in Table 1 and Fig 1. Complete data about each fetus is provided in Supporting information.

## Hemodynamic variations

FHR and MAP were normal at baseline in both groups. In control group, there were no significant changes in FHR or MAP. In the LPS group, FHR was significantly higher at H5 (196 bpm [182; 209], p = 0.01) and H6 (203 bpm [187; 223], p = 0.01) compared with baseline (173 bpm [170; 178]) and MAP did not significantly decrease at H6 (41.0 mmHg [40.0; 42.2], p = 0.07) compared with baseline (45.5 mmHg [43.7; 48.5]).

FHR was significantly higher at H6 in LPS group (203 bpm [187; 223]) compared with control group (181 bpm [177; 188], p = 0.02). MAP was significantly lower at H6 in LPS group (41.0 mmHg [40.0; 42.2]) compared with control group (47.0 mmHg [43.0; 55.5], p = 0.04).

## Blood sample parameters

pH and lactate were normal at baseline in both groups. In the LPS group, pH was significantly lower than baseline (H0 = 7,39 [7.39; 7.40]) from H1 to H6 (H1 = 7.39 [7.37; 7.39], p = 0.04; H6 = 7.33 [7.30; 7.35], p = 0.01). Lactate were significantly higher than baseline (H0 = 2.19 mmol/L [1.99; 2.5]) from H2 to H6 (H2 = 3.16 mmol/L [2.69; 3.85], p = 0.01; H6 = 5.65 mmol/L [4.88; 8.42], p = 0.01).

**Table 1. Hemodynamic, blood gas and biochemical measures in LPS and control groups.**

| | | H0 | H1 | H2 | H3 | H4 | H5 | H6 | p(1) |
|---|---|---|---|---|---|---|---|---|---|
| **FHR bpm** | LPS | 173(170;178) | 166(159;180) | 174(172;180) | 161(155;168) | 181(165;191) | **196(182;209)*** | **203(187;223)*** | <0.001 |
| | Control | 175(173;179) | 178(169;205) | 178(172;182) | 176(172;178) | 182(175;185) | 177(172;196) | 181(177;188) | 0.46 |
| | p(2) | 0.39 | 0.23 | 0.53 | 0.15 | 0.53 | 0.09 | **0.02** | |
| **MAP mmHg** | LPS | 45.5(43.7;48.5) | 49,0(47,0;51.5) | 47,0(45.2;49.2) | 44.5(43,0;47,0) | 46,0(44.7;46.2) | 42,0(41,0;45.2) | 41,0(40,0;42.2) | **0.01** |
| | Control | 48(43;55.5) | 47(43.5;56) | 45(44;53) | 43(42;52) | 48(42.5;50) | 43(40.5;59) | 47(43;55.5) | 0.84 |
| | p(2) | 0.90 | 0.91 | 1 | 1 | 0.48 | 0.48 | **0.04** | |
| **pH** | LPS | 7.39(7.38;7.40) | **7.39 (7.37;7.39)*** | **7.35 (7.32;7.38)*** | **7.32 (7.3;7.35)*** | **7.33 (7.29;7.36)*** | **7.33 (7.29;7.36)*** | **7.33 (7.30;7.35)*** | <0.001 |
| | Control | 7.41(7.39;7.42) | 7.41(7.4;7.43) | 7.41(7.39;7.42) | 7.4(7.4;7.42) | 7.41(7.39;7.42) | 7.4(7.39;7.41) | 7.4(7.38;7.41) | 0.06 |
| | p(2) | 0.26 | **0.01** | **0.01** | **0.01** | **0.01** | **0.01** | **0.01** | |
| **PCO2 mmHg** | LPS | 47.6(45.7;49.1) | 49.3(46.6;50.9) | 51.7(49,0;52.5) | **54.4 (51.0;56.8) *** | **52.3 (48.4;54.4) *** | **52.4 (50.5;55.8) *** | **51.8 (50,0;56.9) *** | <0.001 |
| | Control | 48.1(45.4;50.3) | 48(45.5;50.4) | 48.3(45.7;50.8) | 47.8(45.8;49.6) | 48.1(47.4;49.5) | 48.3(46.9;50.1) | 47.8(47.2;50.6) | 0.071 |
| | p(2) | 0.86 | 0.72 | 0.32 | **0.01** | 0.07 | 0.05 | 0.14 | |
| **PO2 mmHg** | LPS | 17,0(15.7;21.2) | 19,0(17.5;20.2) | 19.5(17.5;20.5) | 16,0(15.2;19,0) | 16,0(14,0;18,0) | 16.5(13.7;17.2) | 16.5(15,0;18.2) | <0.001 |
| | Control | 19(16.5;21.5) | 19(17;19.5) | 19(18;21) | 19(18;21) | **18 (16.5;20)*** | **19 (17.5;20.5)*** | **18 (16;19.5)*** | 0.48 |
| | p(2) | 0.48 | 0.60 | 0.81 | 0.06 | 0.19 | **0.03** | 0.23 | |
| **Base Excess** | LPS | 3.5(3,00;5.5) | 3.5(2,00;5.25) | **1,00 (-1,0;3.5)*** | **1,00 (0.2;2.25)*** | **0.5 (-0.2;2.25)*** | **1.5 (-1,0;2.75)*** | **1,00 (-0.2;3.5)*** | <0.001 |
| | Control | 6(5;6.5) | 6(6;6.5) | 6(4.5;6.5) | 6(5;6.5) | 5(5;6.5) | 6(5;6.5) | 5(4.5;6.5) | 0.48 |
| | p(2) | 0.21 | 0.07 | **0.02** | 0.11 | **0.01** | **0.02** | **0.03** | |
| **HCO3 mmol/L** | LPS | 27.9(27.4;29.6) | 28.2(26.8;29.9) | **26.5 (25.2;28.2)*** | **26.3 (25.6;27.9)** | 25.9 (25.2;26.7) | **26.1 (25.1;28.0)*** | **26.5 (25.1;28.7)*** | **0.02** |
| | Control | 30.2(29.0;30.8) | 30.2(30.1;30.4) | 30.2(28.5;30.7) | 29.8(29.0;30.7) | 29.8(29.3;30.6) | 30.1(29.2;30.7) | 29(28.6;30.7) | 0.48 |
| | p(2) | 0.33 | 0.14 | **0.02** | 0.20 | **0.02** | 0.05 | 0.05 | |
| **Lactate mmol/L** | LPS | 2.19(1.99;2.5) | 2.44(2.34;2.62) | **3.16 (2.69;3.85)*** | **4.44 (3.81;4.94)*** | **4.58 (4.25;6.34)*** | **5.29 (4.84;7.43)*** | **5.65 (4.88;8.42)*** | <0.001 |
| | Control | 2.55(2.46;2.73) | 2.43(2.14;2.59) | 2.35(1.98;2.53) | **2.00 (1.90;2.22)*** | **2.12 (1.79;2.27)*** | **2.1 (1.80;2.32)*** | **2.09 (1.87;2.29)*** | <0.001 |
| | p(2) | 0.24 | 0.95 | **<0.01** | **<0.01** | **<0.01** | **<0.01** | **<0.01** | |
| **Il6 pg/ml** | LPS | <7.8(7.8;7.8) | **54.1(21.7;79.2)** | **1062 (508.;1419)** | **3166 (2789;4207)** | **3330 (2238;3928)** | **3783 (2078;4203)** | **3340 (1970;3801)** | <0.001 |
| | Control | <7.8(7.8;7.8) | <7.8 | <7.8 | <7.8 | <7.8 | <7.8( | <7.8 | 1 |
| | p(2) | 39 | **<0.01** | **<0.01** | **<0.01** | **<0.01** | **<0.01** | **<0.01** | |

Data are presented on median (interquartile range).

Statistical analysis: (1) Comparison from H1 to H6 compared with stability (H0): Friedman nonparametric test; p < 0.05 was significant

* *Wilcoxon test*: *P < 0.05 was significant*.

(2) Comparison between LPS and control group: Mann-Whitney test: P < 0.05 was significant.

FHR = fetal heart rate, MAP = mean arterial pressure, IL6 = interleukin 6.

PO2 was significantly lower than baseline (17.0 mmHg [15.7; 21.2]) at H4 (16.0 mmHg [14.0; 18.0], p = 0.05), H5 (16.5 mmHg [13.7; 17.2], p = 0.042) and H6 (16.5 mmHg [15.0; 18.2], p = 0.035). PCO2 was significantly higher than baseline (H0 = 47.6 mmHg [45.7; 49.1]) from H2 to H6, (H2 = 51.7 mmHg [49.0; 52.5]; H6 = 51.8 mmHg [50.0; 56.9]), p = 0.01). IL-6 was significantly higher than baseline (H0<7.8 pg/ml) from H1 to H6 (H1 = 54.1 pg/ml [21.7; 79.2]; H6 = 3340 pg/ml [1970; 3801], p = 0.01).

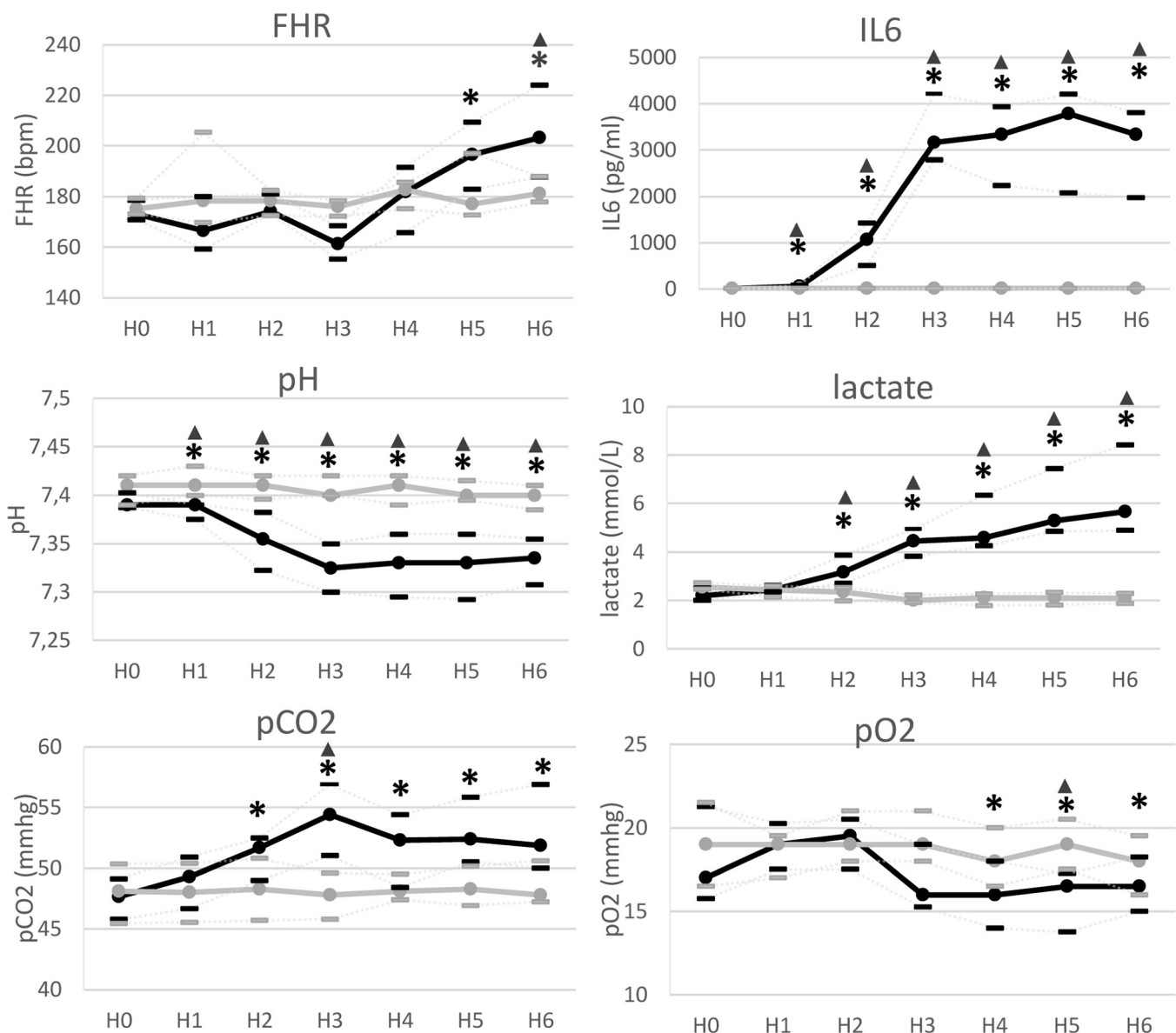

**Fig 1. Evolution of Hemodynamic, blood gas and biochemical parameters in LPS group and control group.** Black LPS group (n = 8); Grey control group (n = 7) at baseline (H0), H1, H2, H3, H4, H5, H6 after LPS (LPS group) or saline injection (control group). FHR = fetal heart rate, IL-6 = interleukin 6. Data are presented on median with interquartile rang. Comparisons between stability phase and H1 to H6 were performed, in LPS group and control group, using a Wilcoxon test if nonparametric Friedmans' test found a statistical significance. * = Statistical significance was assumed for p < 0.05 in comparison to baseline in LPS group. No significantly change was found in control group. Comparisons between LPS and control group were performed using a Mann-Whitney test. ▲ = Statistical significance was assumed for p < 0.05 between LPS and control groups.

In the control group, there were no significant changes in pH, pO2, pCO2, or IL-6. Lactate was significantly lower than baseline (2.55 mmol/L [2.46; 2.73]) from H3 (2.00 mmol/L [1.90; 2.22], p = 0.016) to H6 (2.09 mmol/L [1.87; 2.29], p = 0.016).

In LPS group, compared to control group, pH was significantly lower from H1 to H6 and lactate higher from H2 to H6. At H3, pO2 was significantly lower (p = 0.03) and pCo2 higher (p < 0.01). IL6 was significantly higher from H1 to H6.

### HRV analysis

HRV measures in LPS group from H1 to H6 compared with stability (H0) are shown in Table 2 and Fig 2. In comparison to baseline, five HRV measures changed significantly at least once during the 6 h after LPS injection.

SDNN was significantly higher than baseline (H0 = 8.07 ms [6.11; 10.0]) from H2 to H4 (H2 = 13.8 ms [7.98; 18.0], p < 0.01, H3 = 15.7 ms [12.6; 29.3], p < 0.01 and H4 = 15.3 ms [10.6; 17.4], p = 0.04). DFA α1 was significantly lower than baseline (H0 = 7.49 [5.92; 9.32]) at H2, H3, H4 and H6 (H2 = 0.76 [−1.2; 5.74], p < 0.01, H3 = 0.42[-3.1; 2.44], p < 0.01, H4 = 0.08 [−1.0; 3.46], p < 0.01 and H6 = 4.62 [0.82; 6.12], p = 0.04). DFA α2 was significantly lower than baseline (H0 = 5.97 [4.11; 8.00]) from H2 to H4, (H2 = 0.93 [−1.7; 4.55], p < 0.01, H3 = -0.7 [-2.1; 1.73], p = 0.02 and H4 = 0.24 [−0.8; 2.19], p = 0.02). SD2 was significantly higher than baseline (H0 = 0.98 [0.79; 1.20]) from H2 to H4 (H2 = 1.81 [1.01; 2.35], p < 0.01, H3 = 2.01[1.70; 2.91], p < 0.01 and H4 = 2.04 [1.33; 2.33], p < 0.01). LTV was significantly higher than baseline (H0 = 31.0 ms [25.0; 37.0]) from H2 to H4 (H2 = 49.3 ms [32.2; 63.3], p = 0.02, H3 = 56,6 ms [51,9; 75,2], p < 0.01 and H4 = 58.1 ms [40.8; 64.1], p < 0.01). No significant variation was found in any of the other HRV measures. No significant variation was found in any HRV measure in the control group in comparison to baseline.

Comparison between LPS and control groups are shown in Table 2 and Fig 2. Five HRV measures significantly differed at least once during the 6 h after LPS injection in comparison to control group. DFA alpha1 was significantly lower at H3 (0.42 [-3.1; 2.44] vs 4.44 [3.32; 6.35], p < 0.01) in LPS group compared to control group. DFA alpha 2 was significantly lower at H3 (-0.7 [-2.1; 1.73] vs 3.31 [1.55; 3.86], p = 0.02) in LPS group compared to control group. ApEn was significantly lower at H3 (0.49 [0.45; 0.55] vs 0.57 [5.47; 5.82], p = 0.02) in LPS group compared to control group. LTV was significantly higher at H2 (49.3 [32.2; 63.3] vs 33.0 [26.8; 42.7], p = 0.04) and H4 (58.1 [40.8; 64.1] vs 41.2 [34.8; 48.3], p < 0.01) in LPS group, compared to control group. SDNN was significantly higher at H6 (9.50 [8.80; 13.0] vs 13.5 [11.1; 16.3], p = 0.02) in LPS group compared to control group.

## Discussion

In this near-term fetal sheep experimental FIRS model, hemodynamic, gasometric, and HRV parameters changed after LPS injection. FIRS was associated with increased FHR, decreased pH, and increased lactate. Among the 14 HRV indices analyzed, five changed significantly at least once after LPS injection in comparison to baseline. These five indices were: SDNN, DFA α1, DFA α2, LTV, and SD2, all of which changed significantly from H2 to H4.

Among these five indices, DFA α1, DFA α2, LTV and SD2 demonstrated significant differences between LPS and control groups after LPS injection (between H2 and H4). ApEn was significantly lower at H3 in LPS group in comparison to control group.

FIRS is a systemic inflammation with elevated fetal IL-6 [25]. Elevated IL-6 in the LPS group 1 h after LPS injection confirmed successful FIRS model creation. This model was previously used by Durosier et al., who administered intravenous LPS to 10 fetal sheep, inducing FIRS without shock or cardiovascular decompensation [17]. These investigators also found septicemia 3 h after LPS injection, with a slight blood pressure drop, FHR increase, mild hypoxia, and IL-6 rise [17].

In our model, pH decreased starting at H1 and lactate increased starting at H2. PCO2 increased from H2, and PO2 decreased from H4. Placental vascular resistance can explain some of these gas changes. Several mechanisms can cause increased placental resistance [26]. First, secretion of endothelin 1, a potent vasoconstrictive peptide. Second, formation of placental edema caused by increase permeability. Third, increased flow to the brain, heart, and

**Table 2. Heart rate variability measures in LPS and control groups.**

| | | H0 | H1 | H2 | H3 | H4 | H5 | H6 | p(1) |
|---|---|---|---|---|---|---|---|---|---|
| **SDNN (ms)** | LPS | 8.07(6.11;10.0) | 8.82(7.14;12.3) | **13.8 (7.98;18.0)*** | **15.7 (12.6;22.3)*** | **15.3 (10.6;17.4)*** | 10.6(10.0;15.8) | 9.50(8.80;13.0) | **<0.001** |
| | Control | 12.0(9.55;13.9) | 12.3(10.2;13.5) | 9.26(8.68;9.51) | 11.6(9.71;13.1) | 10.9(9.77;12.4) | 10.1(8.98;13.5) | 13.5(11.1;16.3) | 0.44 |
| | p(2) | 0.12 | 0.33 | 0.28 | 0.09 | 0.23 | 0.69 | 0.02 | |
| **RMSSD (ms)** | LPS | 6.22(3.98;8.60) | 6.51(4.94;8.98) | 9.06(5.24;11.9) | 11.4(6.92;14.4) | 8.22(6.22;11.8) | 8.57(6.92;13.5) | 8.20(5.34;10.2) | 0.13 |
| | Control | 6.40(6.03;12.8) | 7.27(6.22;9.14) | 6.43(5.63;8.37) | 8.73(5.67;10.6) | 8.53(5.79;9.89) | 7.32(6.11;10.2) | 13.2(7.11;19.5) | 0.36 |
| | p(2) | 0.23 | 0.61 | 0.86 | 0.18 | 0.69 | 0.68 | 0.28 | |
| **LF (dB)** | LPS | 5.65(4.76;6.95) | 5.05(4.69;6.06) | 5.10(4.91;5.52) | 4.66(4.13;5.23) | 4.98(4.39;5.23) | 5.50(5.18;5.71) | 5.59(4.82;5.94) | 0.80 |
| | Control | 1.86(1.58;1.99) | 1.71(1.67;1.82) | 1.82(1.70;1.91) | 1.71(1.67;1.87) | 1.89(1.65;1.94) | 1.79(1.69;1.82) | 1.76(1.58;1.94) | 0.90 |
| | p(2) | 0.61 | 0.23 | 0.33 | 0.69 | 0.77 | 1 | 0.77 | |
| **HF (dB)** | LPS | 5.97(4.11;8.00) | 4.78(1.61;5.39) | 0.93(-1.7;4.55) | -0.7(-2.1;1.73) | 0.24(-0.8;2.19) | 3.15(1.08;3.89) | 2.97(0.42;4.97) | 0.22 |
| | Control | 5.31(4.93;6.42) | 5.82(5.46;6.79) | 5.54(4.59;6.07) | 6.10(4.00;7.02) | 5.84(5.20;6.35) | 6.17(4.28;6.17) | 6.00(5.68;7.72) | 0.87 |
| | p(2) | 0.86 | 0.56 | 0.53 | 0.12 | 0.69 | 0.95 | 0.53 | |
| **LFHF ratio** | LPS | 3.92(2.26;4.33) | 3.45(3.15;4.19) | 3.73(3.07;4.50) | 4.23(3.42;5.01) | 4.13(3.55;6.26) | 3.71(3.12;3.91) | 3.92(3.09;4.41) | 0.32 |
| | Control | 4.10(2.65;5.13) | 3.94(3.45;4.15) | 4.08(2.89;0.47) | 3.42(2.82;4.77) | 4.01(2.95;4.41) | 3.56(3.02;3.93) | 2.34(1.88;3.14) | 0.21 |
| | p(2) | 0.61 | 0.95 | 1 | 0.61 | 0.69 | 0.95 | 0.07 | |
| **sd1** | LPS | 4.39(2.81;6.08) | 4.59(3.49;6.35) | 6.40(3.69;8.43) | 8.10(4.89;1.01) | 5.80(4.39;8.36) | 6.05(4.88;9.54) | 5.79(3.77;7.24) | 0.13 |
| | Control | 4.52(4.25;9.07) | 5.13(4.39;6.45) | 4.54(3.97;5.91) | 6.17(4.00;7.54) | 6.02(4.09;6.99) | 5.17(4.31;7.21) | 9.36(5.02;1.38) | 0.36 |
| | p(2) | 0.23 | 0.61 | 0.86 | 0.18 | 0.69 | 0.61 | 0.28 | |
| **sd2 (E-1)** | LPS | 0.98(0.79;1.20) | 1.09(0.92;1.35) | **1.81 (1.01;2.35)*** | **2.01 (1.70;2.91)*** | **2.04 (1.33;2.33)*** | 1.34(1.25;1.91) | 1.16(1.08;1.67) | **0.01** |
| | Control | 1.42(1.07;1.86) | 1.51(1.20;1.64) | 1.16(1.09;1.21) | 1.47(1.20;1.56) | 1.43(1.25;1.55) | 1.28(1.15;1.73) | 1.59(1.45;1.78) | 0.47 |
| | p(2) | 0.18 | 0.33 | 0.23 | **0.04** | 0.23 | 0.77 | 0.28 | |
| **ratioSD** | LPS | 6.65(4.34;7.46) | 5.41(4.33;6.00) | 4.76(4.16;4.97) | 4.38(4.02;4.77) | 4.14(3.70;4.82) | 4.96(4.38;6.51) | 5.48(4.75;6.13) | 0.03 |
| | Control | 4.71(3.33;6.75) | 4.78(3.66;5.83) | 4.74(4.40;5.46) | 5.56(4.07;5.74) | 4.61(3.90;5.05) | 4.60(4.36;5.18) | 5.40(4.60;6.44) | 0.91 |
| | p(2) | 0.46 | 0.61 | 0.61 | 0.46 | 0.61 | 0.46 | 1 | |
| **DFAα1 (E-18)** | LPS | 7.49(5.92;9.32) | 6.10(3.23;7.25) | **0.76 (-1.2;5.74)*** | **0.42 (-3.1;2.44)*** | **0.08 (-1.0;3.46)*** | 3.92(1.89;4.60) | 4.62(0.82;6.12) | **<0.001** |
| | Control | 3.53(3.59;8.67) | 3.38(1.35;6.13) | 5.73(4.13;6.52) | 4.44(3.32;6.35) | 3.69(2.90;4.23) | 4.31(2.75;5.33) | 3.09(2.51;4.14) | 0.54 |
| | p(2) | 0.23 | 0.46 | 0.09 | **0.01** | 0.15 | 1 | 0.53 | |
| **DFAα2 (E-18)** | LPS | 5.97(4.11;8.00) | 4.78(1.61;5.39) | **0.93 (-1.7;4.55)*** | -0.7 (-2.1;1.73)* | **0.24 (-0.8;2.19)*** | 3.15(1.08;3.89) | 2.97(0.42;4.97) | **<0.001** |
| | Control | 2.37(2.26;4.95) | 3.57(7.67;5.39) | 4.82(3.24;5.40) | 3.31(1.55;3.86) | 2.33(1.03;3.68) | 3.37(2.75;3.88) | 1.93(7.98;3.73) | 0.61 |
| | p(2) | 0.09 | 0.69 | 0.07 | **0.02** | 0.07 | 0.77 | 0.69 | |
| **ApEn** | LPS | 0.61(0.55;0.63) | 0.59(0.53;0.62) | 0.54(0.48;0.62) | 0.49(0.45;0.55) | 0.53(0.49;0.57) | 0.59(0.54;0.59) | 0.55(0.47;0.59) | 0.26 |
| | Control | 0.55(5.17;5.74) | 0.54(5.22;5.61) | 0.57(5.66;5.93) | 0.57(5.47;5.82) | 0.56(5.28;5.67) | 0.54(5.14;5.76) | 0.53(5.16;5.58) | 0.35 |
| | p(2) | 0.15 | 0.18 | 0.86 | **0.02** | 0.28 | 0.39 | 0.46 | |
| **STV** | LPS | 3.01(2.36;3.40) | 3.54(3.17;5.05) | 4.64(3.09;6.43) | 5.44(5.28;6.90) | 4.53(2.35;5.54) | 3.89(2.66;6.33) | 4.46(3.85;5.16) | 0.30 |
| | Control | 3,36(3.01;4.58) | 3.10(2.78;4.05) | 3.07(2.74;3.85) | 3.70(3.04;3.99) | 4.63(3.46;4.81) | 4.06(3.44;4.55) | 3.89(3.54;4.23) | 0.72 |
| | p(2) | 0.30 | 0.24 | 0.9 | 0.6 | 0.58 | 0.93 | 0.39 | |
| **LTV** | LPS | 31.0(25.0;37.0) | 37.9(32.5;50.2) | **49.3 (32.2;63.3)*** | **56.6 (51.9;75.2)*** | **58.1 (40.8;64.1)*** | 41.8(27.8;59.3) | 42.8(39.4;56.8) | **<0.001** |
| | Control | 34.5(29.5;47.6) | 34.8(30.5;42.9) | 33.0(26.8;42.7) | 41.2(35.7;45.3) | 41.2(34.8;48.3) | 35.8(34.5;47.5) | 43.2(35.8;45.1) | 0.27 |
| | p(2) | 0.48 | 0.39 | **0.04** | 0.6 | **0.01** | 0.48 | 0.81 | |

*(Continued)*

**Table 2.** (Continued)

| | | H0 | H1 | H2 | H3 | H4 | H5 | H6 | p(1) |
|---|---|---|---|---|---|---|---|---|---|
| **FSI** | LPS | 52.1(48.5;53.6) | 46.7(44.9;53.2) | 45.2(44.7;48.2) | 48.0(43.7;51.1) | 43.7(42.1;45.4) | 45.6(42.3;50.0) | 46.6(43.0;49.9) | 0.60 |
| | Control | 49.6(45.7;53.3) | 48.3(47.4;55.2) | 49.3(40.1;51.7) | 44.0(39.6;55.6) | 54.2(47.9;56.9) | 47.5(42.9;49.8) | 50.1(45.5;66.1) | 0.54 |
| | p(2) | 0.83 | 0.53 | 1.00 | 0.77 | 0.05 | 0.69 | 0.53 | |

Data are presented on median (interquartile range).

Statistical analysis: (1) Comparison from H1 to H6 compared with stability (H0): Friedman nonparametric test; $p < 0.05$ was significant

* Wilcoxon test: $P < 0.05$ was significant.

(2) Comparison between LPS and control group: Mann-Whitney test: $P < 0.05$ was significant.

SDNN = standard deviation of normal to normal, R-R intervals, RMSSD = root mean square of successive differences, DFA = detrended fluctuation analysis,

SD1 = Standard Deviation 1, SD2 = Standard Deviation 2, FSI = fetal stress index, ApEn = Approximate Entropy,.

STV = short-term variability, LTV = long-term variability, LF = low frequencies, HF = high frequencies.

adrenals at the expense of placental flow [26]. FHR elevation that appeared at H5 can be explained by secretion of catecholamines induced by LPS injection [26]. No significant MAP modification was found, but there was a decreasing trend at H5 that could be explained by endothelial dysfunction caused by inflammation. This hemodynamic response and increased IL-6 from H1 after LPS injection validate this FIRS model, without shock or cardiovascular decompensation. The hemodynamic and gasometric responses to LPS injection herein were similar to other studies after intravenous LPS in near-term fetal sheep [17,27].

In fetal sheep, LPS bolus administration was inconstantly associated with an increase of HRV [17,27–31]. These studies used different HRV analyses, with different models of FIRS (acute, chronic), by different LPS injection modes (intraamniotic et intravenous) at different gestational age. Only two teams have studied HRV changes after intravenous LPS injection in the near-term fetus. Blad et al. studied ECG and HRV changes in preterm and near-term fetal sheep following LPS exposure, finding no HRV changes or tachycardia after one intravenous LPS injection [27]. However, only one global HRV analysis was conducted and the dose (100 ng/kg) and LPS (Sigma O55:B5) differed from those used herein. Durosier et al. used a multi-dimensional analysis of complementary HRV measures from different signal-analytic domains, allowing detection of FIRS [17]. That group selected five HRV indices but the parameter characteristics were not separately described. Our aim herein was to study each of the main HRV indices during modeled FIRS.

SDNN, DFA α1, DFA α2, SD2, LTV, and APEN are altered after LPS injection in the sheep fetus, even before fetal tachycardia onset. It has been shown that HRV increase can be concomitant with tachycardia [27,29,31]. In these studies, fetuses were premature, and fetal reactions were stronger with strong hypotension. However, discrepancy between the occurrence of tachycardia and changes in HRV parameters has already been reported in two studies in fetal sheep at term after LPS injection [17,30]. First, Durosier et al found that discrimination between LPS-injected animals and control group using HRV occurs between 2 h and 3 h post injection. Whereas FHR was significantly higher in LPS group only at H6. Second, Kyozuka et al., who studied short-term variability changes after LPS infusion into the amniotic cavity, found significative change in short-term variability, with no tachycardia, at 6, 4, and 3 h before intrauterine fetal death [30].

HRV indices can reflect sympathetic, parasympathetic, or global autonomic activities [19]. Activation of the adrenergic system in sepsis is critical for initiating a physiologic response to pathogens but can become detrimental in excess [13]. The cholinergic anti-inflammatory pathway can regulate the inflammatory response via the vagus nerve, providing negative feedback on systemic inflammatory cytokine levels [11].

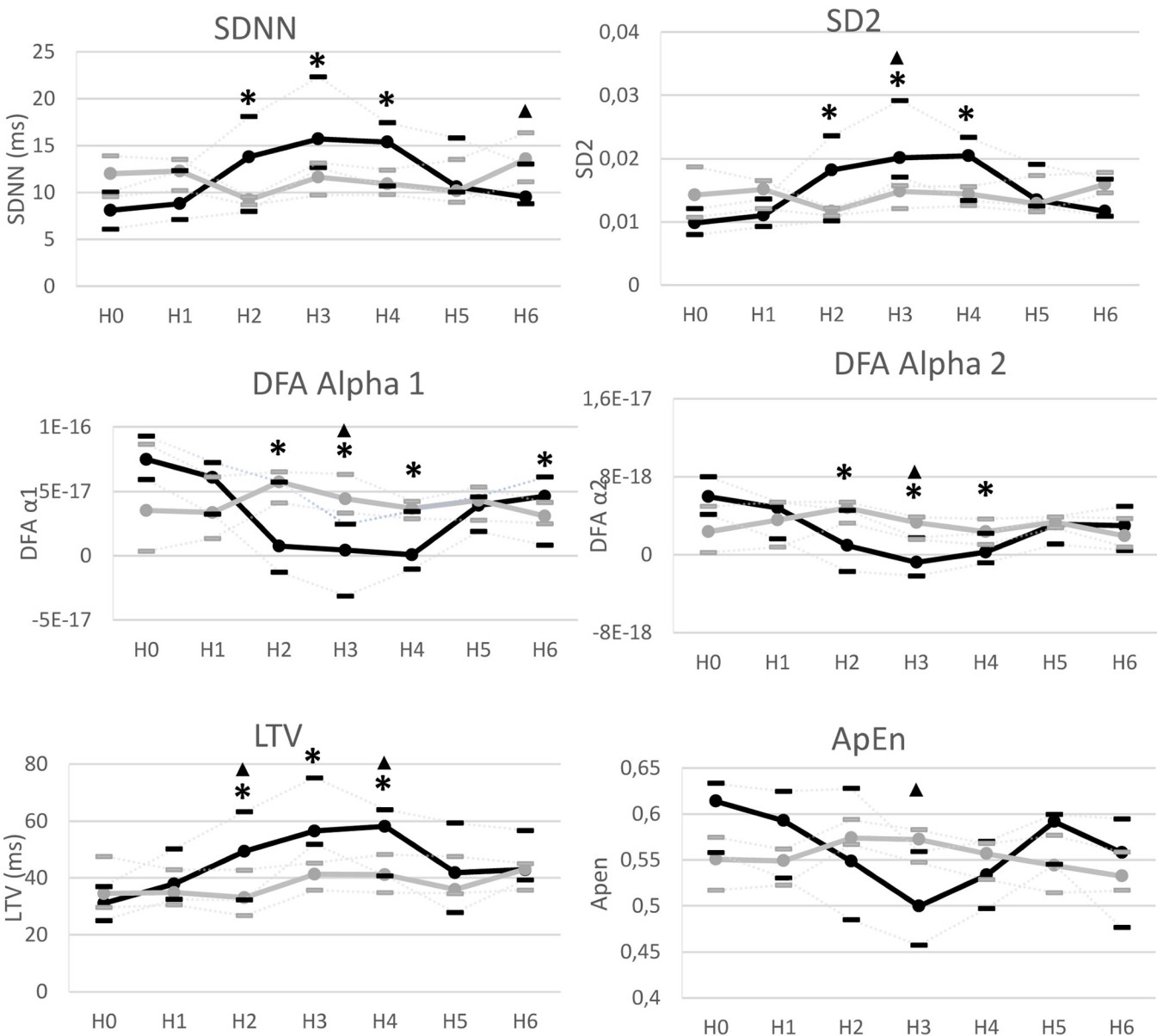

**Fig 2. Evolution of Heart rate variability measures in LPS group and control group.** *Black LPS group (n = 8); Grey control group (n = 7) at baseline (H0), H1, H2, H3, H4, H5, H6 after LPS (LPS group) or saline injection (control group). SDNN = standard deviation of normal to normal, DFA = detrended fluctuation analysis, SD2 = Standard Deviation 2, LTV = long term variability, ApEn = Approximate Entropy Data are presented on median with interquartile rang. Comparisons between stability phase and H1 to H6 were performed, in LPS group and control group, using a Wilcoxon test if nonparametric Friedmans' test found a statistical significance. * = Statistical significance was assumed for p < 0.05 in comparison to baseline in LPS group. No significantly change was found in control group. Comparisons between LPS and control group were performed using a Mann-Whitney test. ▲ = Statistical significance was assumed for p < 0.05 between LPS and control groups.*

FSI, RMSSD, and HF has been associated with parasympathetic activity [19,22]. That we failed to find any fluctuations in these indices after LPS injection, could suggest an absence of change in parasympathetic activity. [11]. After LPS injection, SDNN increased from H2 to H4 in comparison to baseline. Both sympathetic activity and parasympathetic activity contribute to SDNN. This increase of SDNN with no parasympathetic activation could suggest a sympathetic activation. However, a sympathetic activation with no parasympathetic activation should

have increased the LF/HF ratio, what we did not highlight. It is also unexpected that the onset of tachycardia from H5 was associated with a return to baseline of HRV markers. It has already been shown that initial increase in HRV can be followed by suppression of time-domain measures [27,28,29,31]. However, with the onset of tachycardia, we might have expected an increase of indices usually associated with sympathetic activity. We have no pathophysiological explanation for this discrepancy. Altogether, inhibition or activation of autonomic activities cannot be clearly established by our HRV indices changes. It must be noted that each parameter does not strictly reflect direct sympathetic or parasympathetic individual changes [32] and there is a substantial overlap in the sympathetic and parasympathetic spectrum [33].

Early HRV changes also appear to be mediated by a loss of signal complexity. After LPS injection, DFA α1 and α2 decreased from H2 to H4 in comparison to baseline and was lower at H3 in comparison to control group. ApEn was lower at H3 in comparison to control group. DFA α1, DFA α2 and ApEn are nonlinear measurements [19]. Nonlinear measurements provide information about the unpredictability of a time series, which results from the complexity of the mechanisms that regulate HRV. Stressors like infection are known to depress some nonlinear measurements [19]. DFA α1 (window width: $4 \leq n \leq 16$ beats) was previously applied to clinical cardiovascular risk assessment, prognosis, and mortality prediction [21]. Applying DFA to human adults, Brown et al. found that a loss of complexity in HRV with DFA changes can predict, during severe sepsis or septic shock, early resuscitation success [34]. To our knowledge, only Durosier et al. have studied DFA and entropy in a fetal sheep model of infection. In their study, DFA and entropy indices were tested but not selected to create a HRV composite measure to monitor the inflammatory response. However, the HRV measures selected increased and correlation with inflammation in the three days after initial LPS injection. Herein, these findings support DFA and ApEn, as nonlinears measures, as interesting tools for detecting loss of signal complexity early in acute fetal infection.

Significant changes in SDNN, DFA α1, DFA α2, SD2, LTV and ApEn were detected before tachycardia in our model. These changes seem related to sympathetic activation and a loss a signal complexity. Thus, these indices may be valuable for early detection of acute fetal infection. FIRS is associated with severe morbidity and mortality, and new tools for its detection are needed [3–7]. HRV may be promising for early detection of fetal infection. To date, no study has evaluated HRV indices during FIRS in the human fetus. Few studies have evaluated fetal HRV in FIRS with near-term fetal sheep [17,27,30], which share many similarities with human gestation [35]. Our study is the first to present characteristics of 14 changed indices after LPS injection. With the improvement of fetal ECG recording systems, and thus R–R signal quality, HRV indices may soon be feasible for detecting fetal disturbances like FIRS.

This experiment was not without limitations. First, it focused specifically on acute onset FIRS in the near-term fetus and the model does not therefore reflect the entire range of fetal infections, which may be subacute or chronic with differing intensity. Further, we focused only the fetal system; concomitant infections of the placenta, amniotic fluid, or maternal system may alter these indices in different ways. For example Kyozuka et al., who studied short-term variability changes after LPS infusion into the amniotic cavity, found different results [30]. Specifically, no significant changes were found during the first 24 h post-injection, while short-term variability at 6, 4, and 3 h before intrauterine fetal death increased significantly. Nevertheless, only one HRV analysis was used in that study: short-term variability defined as the average of differences between two intervals. The second study limitation is that although we used an animal model of human gestation, generalizability of findings and their applications to human fetuses must be carefully established. Third, due to small number of fetal sheep, it was not possible to research eventual differences in HRV modifications between singletons and twins.

## Conclusion

SDNN, DFA α1, DFA α2, SD2, LTV, and APEN are altered after LPS injection in the sheep fetus, even before fetal tachycardia onset. These changes appear to be mediated by an increase of global variability and a loss of signal complexity. Thus, these HRV indices may facilitate early detection of acute FIRS.

## Supporting information

**S1 Table. Hemodynamic, blood gas and biochemical measures in LPS and control groups.** MAP = mean arterial pressure, IL6 = interleukin 6. (ODT)

**S2 Table. Heart rate variability measures in LPS and control groups.** *SDNN = standard deviation of normal to normal, R-R intervals, RMSSD = root mean square of successive differences, DFA = detrended fluctuation analysis, SD1 = Standard Deviation 1, SD2 = Standard Deviation 2, FSI = fetal stress index, ApEn = Approximate Entropy, STV = short-term variability, LTV = long-term variability, LF = low frequencies, HF = high frequencies.* (ODT)

## Acknowledgments

We thank Capucine Besengez and all the staff of the Research Experimental Department of University Lille North of France for their veterinary care and their expert assistance with sheep surgery.

## Author Contributions

**Conceptualization:** Charles Garabedian, Laurent Storme, Louise Ghesquière.

**Data curation:** Geoffroy Chevalier.

**Formal analysis:** Anne Wojtanowski, Julien De Jonckheere.

**Investigation:** Geoffroy Chevalier, Jean David Pekar, Delphine Le Hesran, Louis Edouard Galan, Dyuti Sharma.

**Methodology:** Louise Ghesquière.

**Project administration:** Charles Garabedian, Louise Ghesquière.

**Supervision:** Charles Garabedian, Veronique Houfflin-Debarge.

**Writing – original draft:** Geoffroy Chevalier, Charles Garabedian, Louise Ghesquière.

**Writing – review & editing:** Laurent Storme.

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
