## [Decision Letter · Decision Letter 0]

9 Jul 2023

PONE-D-23-13482Early heart rate variability changes during acute fetal inflammatory response syndrome: an experimental study in a fetal sheep modelPLOS ONE

Dear Dr. Chevalier,

Thank you for submitting your manuscript to PLOS ONE. After careful consideration, we feel that it has merit but does not fully meet PLOS ONE’s publication criteria as it currently stands. Therefore, we invite you to submit a revised version of the manuscript that addresses the points raised during the review process. Please submit your revised manuscript by Aug 23 2023 11:59PM. If you will need more time than this to complete your revisions, please reply to this message or contact the journal office at plosone@plos.org. Please include the following items when submitting your revised manuscript:A rebuttal letter that responds to each point raised by the academic editor and reviewer(s). You should upload this letter as a separate file labeled 'Response to Reviewers'.A marked-up copy of your manuscript that highlights changes made to the original version. You should upload this as a separate file labeled 'Revised Manuscript with Track Changes'.An unmarked version of your revised paper without tracked changes. You should upload this as a separate file labeled 'Manuscript'.

We look forward to receiving your revised manuscript.

Kind regards,

Sanjoy Kumer Dey, M.D

Academic Editor

PLOS ONE

3. Please amend the manuscript submission data (via Edit Submission) to include author L. Storme.

Reviewers' comments:

Reviewer's Responses to Questions

**Comments to the Author**

1. Is the manuscript technically sound, and do the data support the conclusions?

Reviewer #1: No

Reviewer #2: Yes

2. Has the statistical analysis been performed appropriately and rigorously? 

Reviewer #1: I Don't Know

Reviewer #2: Yes

3. Have the authors made all data underlying the findings in their manuscript fully available?

Reviewer #1: Yes

Reviewer #2: Yes

4. Is the manuscript presented in an intelligible fashion and written in standard English?

Reviewer #1: Yes

Reviewer #2: Yes

5. Review Comments to the Author

Reviewer #1: You have induced inflammation by administering LPS to near-term fetal sheep and evaluated the associated changes in various FHRV parameters.

While the method itself is very systematic, your discussion of the results is misleading to the reader. Therefore, I anticipate that this paper will be difficult to accept unless significant revisions are made.

Methods

・Why did you not use Sample Entropy to evaluate the regularity and complexity of the time series?

・Please correct the statement "FSI reflects parasympathetic nervous system fluctuations" for the reasons discussed below.

・It is stated as Mann-Withney. Please correct the typo.

Regarding the results

Fewer than half of the cases in the Control and LPS groups were singletons. It would be necessary to state whether there were twins or more. It is also necessary to mention whether there was any difference in response to LPS between twins and singletons, as has been the case in previous reports.

Is the p-value of H2 for FSI a misstatement?

Regarding the discussion

I request a fundamental change. You state that FSI, RMSSD, and HF correlate with parasympathetic activity, and that the lack of variation in these indices after LPS injection indicates no change in parasympathetic activity. Besides, you considers that both sympathetic and parasympathetic activity contribute to SDNN, and that the increase in SDNN in the absence of parasympathetic activation indicates that sympathetic activation has occurred.

Regarding the fetal sheep HRV parameter, there is indeed substantial overlap in the sympathetic and parasympathetic spectrum, although HRV itself is predominantly ANS (PMID: 24014809).

It must be noted that each parameter does not strictly reflect direct sympathetic or parasympathetic individual changes (PMID: 25063795).

Also, the presence of tachycardia in H5 cannot be ruled out as a result of the production of catecholamines associated with parasympathetic activation. In fact, the timing of the tachycardia onset does not coincide with the SDNN elevation. How do you interpret this? It is also a significant leap of logic to say that significant changes in SDNN, DFA α1, DFA α2, SD2, LTV, and ApEn are detected prior to tachycardia and that these changes seem to be related to sympathetic activation and loss of signal complexity.

Why was there no change in LF/HF, which was shown to represent an imbalance between parasympathetic and sympathetic nerves? Consideration of this part of the question is lacking.

Lear CA et al. have shown that after repeated hypoxia, sympathetic control of FHRV is rapidly suppressed and parasympathetic control becomes essentially fully dominant (PMID: 32491938, 25864517, 32877241). While intermittent hypoxic exposure also elicits tachycardia, suggesting parasympathetic involvement, if there is clear evidence that a fetus exposed to inflammation becomes sympathetically dominant, it must be stated. If there is no evidence to back it up, then your discussion is a leap too far.

Also, as a fundamental issue, the general view from adult studies is that the relationship between ANS activity and spectral analysis of HRV is obscured at the extremes of physiology, and caution should be exercised in interpreting the results of HRV studies (PMID: 8093124).

Fetal adaptation to inflammatory stress may be an extreme physiological situation that cannot be monitored in adults.

"Quantitation of any variability via whatever technique does not represent an analysis of the complex interactions underlying the variability, but merely a measurement of the resulting phenomenon."(PMID: 16645191), one must be careful in interpreting the results.

Reviewer #2: I am pleased to have the opportunity to review this manuscript. This study examined fetal heart rate variability (HRV) using a preterm fetal inflammation model. The authors reported that five HRV measures changed significantly from H2 to H4, concluding that the measures may be clinically useful for detecting early fetal inflammatory responses. This study was well-designed and yielded insightful results. Given the rarity of physiological studies using large animals, I want to express my respect to the authors for the tremendous importance of their work.

I believe it would be beneficial to revise the Introduction a bit more.

Line66-68: Authors describe “Unfortunately, current understanding of how infection/inflammation affects fetal HRV is limited, in part due to the heterogeneous nature of clinical infections [15].”

While I, as an obstetrician and gynecologist, can very much understand what is being conveyed, I think it would be better to rephrase this statement to reflect that the diagnostic criteria for intrauterine infection or inflammation are not unified across pathology, biochemistry, and clinical judgment. Also, the transition from this statement to lines 69-70 is abrupt. It should be described why a hypothesis was formulated that HRV analysis would be useful for early detection.

I would look forward to the opportunity to review this paper again, should the chance arise.

6. PLOS authors have the option to publish the peer review history of their article (what does this mean?). If published, this will include your full peer review and any attached files.

Reviewer #1: No

Reviewer #2: No

---

## [Author Response · Author response to Decision Letter 0]

6 Aug 2023

Dear Editor,

We would like to thank you for giving us the opportunity to submit a corrected version of our work in your journal.

We also thank the reviewers for the time spent judging our manuscript and the quality of their comments. We have made every effort to respond to each of the comments and believe that the quality of the manuscript is greatly improved by the suggestions made to us.

Corrections appear in highlighted in the manuscript and are detailed in the table in the file "response to reviewers"

We hope this revised article deserves publication in your journal, 

Best regards, 

On the behalf of the co-authors, 

REQUIREMENTS RESPONSES

1. “Please ensure that your manuscript meets PLOS ONE's style requirements, including those for file naming.”

Following items have been corrected:

- Level 1 heading are in Bold type, 18pt 

- Level 2 heading are in Bold type, 16pt

- References have been conformed to Plos one guidelines

- Authors and each affiliation have been corrected

- File naming has been corrected

2. “In your Data Availability statement, you have not specified where the minimal data set underlying the results described in your manuscript can be found.”

 We added Complete data about each fetus is provided in Supporting information.

3. “Please amend the manuscript submission data (via Edit Submission) to include author L. Storme”

 We include him via Edit Submission

Reviewer’s comments:

REVIEW RESPONSE

Reviewer 1:

“I request a fundamental change. You state that FSI, RMSSD, and HF correlate with parasympathetic activity, and that the lack of variation in these indices after LPS injection indicates no change in parasympathetic activity. Besides, you considers that both sympathetic and parasympathetic activity contribute to SDNN, and that the increase in SDNN in the absence of parasympathetic activation indicates that sympathetic activation has occurred.

Regarding the fetal sheep HRV parameter, there is indeed substantial overlap in the sympathetic and parasympathetic spectrum, although HRV itself is predominantly ANS (PMID: 24014809).

It must be noted that each parameter does not strictly reflect direct sympathetic or parasympathetic individual changes (PMID: 25063795).

Also, the presence of tachycardia in H5 cannot be ruled out as a result of the production of catecholamines associated with parasympathetic activation. In fact, the timing of the tachycardia onset does not coincide with the SDNN elevation. How do you interpret this? It is also a significant leap of logic to say that significant changes in SDNN, DFA α1, DFA α2, SD2, LTV, and ApEn are detected prior to tachycardia and that these changes seem to be related to sympathetic activation and loss of signal complexity.

Why was there no change in LF/HF, which was shown to represent an imbalance between parasympathetic and sympathetic nerves? Consideration of this part of the question is lacking.

Lear CA et al. have shown that after repeated hypoxia, sympathetic control of FHRV is rapidly suppressed and parasympathetic control becomes essentially fully dominant (PMID: 32491938, 25864517, 32877241). While intermittent hypoxic exposure also elicits tachycardia, suggesting parasympathetic involvement, if there is clear evidence that a fetus exposed to inflammation becomes sympathetically dominant, it must be stated. If there is no evidence to back it up, then your discussion is a leap too far.”

Also, as a fundamental issue, the general view from adult studies is that the relationship between ANS activity and spectral analysis of HRV is obscured at the extremes of physiology, and caution should be exercised in interpreting the results of HRV studies (PMID: 8093124).

Fetal adaptation to inflammatory stress may be an extreme physiological situation that cannot be monitored in adults.

"Quantitation of any variability via whatever technique does not represent an analysis of the complex interactions underlying the variability, but merely a measurement of the resulting phenomenon."(PMID: 16645191), one must be careful in interpreting the results.”

We thank you for the quality and the explanation of this comments.

We agreed that our results do not allow to clearly conclude to a sympathetic activation. 

We change this part in our discussion and conclusion:

Discussion: l351-372

We have removed: “FSI, RMSSD, and HF are known to be correlated with parasympathetic activity [20,23]. That we failed to find any fluctuations in these indices after LPS injection suggests an absence of change in parasympathetic activity. In contrast, after LPS injection, LTV, SD2, and SDNN increased from H2 to H4 in comparison to baseline. SD2 increased at H3 and LTV increased at H2 and H4 in comparison to control group. SD2 measures short and long-term HRV and is correlated with SDNN [21]. Both sympathetic and parasympathetic activities contribute to SDNN. Because SDNN increases in the absence of parasympathetic activation, we can presume that sympathetic activation occurred. The presence of tachycardia at H5 also supports the presence of sympathetic activation.” 

We have added =>”FSI, RMSSD, and HF has been associated with parasympathetic activity [20,23]. That we failed to find any fluctuations in these indices after LPS injection, could suggest an absence of change in parasympathetic activity. In contrast, after LPS injection, SDNN increased from H2 to H4 in comparison to baseline. Both sympathetic activity and parasympathetic activity contribute to SDNN. This increase of SDNN with no parasympathetic activation could suggest a sympathetic activation. However, a sympathetic activation with no parasympathetic activation should have increased the LF/HF ratio, what we did not highlight. Altogether, inhibition or activation of autonomic activities cannot be clearly established by our HRV indices changes. It must be noted that each parameter does not strictly reflect direct sympathetic or parasympathetic individual changes [32] and there is a substantial overlap in the sympathetic and parasympathetic spectrum [33].”

Conclusion: (Line 411-416): 

We have removed: ”These changes appear related to sympathetic activation and loss of signal complexity”

We have added =>These changes appear to be mediated by an increase of global variability and a loss of signal complexity.

“Please correct the statement "FSI reflects parasympathetic nervous system fluctuations" for the reasons discussed below.” 

Garabedian et al have found that FSI is correlated with parasympathetic activity. In fetal sheep, after intravenous atropine, to inhibit parasympathetic tone, FSI was significantly decreased (1). Garabedian C, Champion C, Servan-

Schreiber E, Butruille L, Aubry E, Sharma D, et al. (2017) A new analysis of heart rate variability in the assessment of fetal parasympathetic activity: An experimental study in a fetal sheep model. PLoS ONE 12(7): e0180653.

We have corrected the sentence: (l160)

In previous experimental studies, we demonstrated that FSI reflects parasympathetic fluctuation and is correlated with acidosis [10,16,17,24,25]. => In previous experimental studies, we demonstrated that FSI was correlated with acidosis and parasympathetic activation [10,16,17,17,24,25].

“Why did you not use Sample Entropy to evaluate the regularity and complexity of the time series?”

To evaluate the regularity and complexity of the time series, we choose Approximate Entropy. The advantage of Sample Entropy is its independence on time-series length; thus it can be computed from shorter HRV records. However, we had a large number of samples within the 64 s window used for HRV analysis (512 samples). This is why we have chosen to focus on Approximate Entropy

“It is stated as Mann-Withney. Please correct the typo” 

Typo has been corrected : =>L182: were performed by Mann-Whitney test.

“Fewer than half of the cases in the Control and LPS groups were singletons. It would be necessary to state whether there were twins or more. It is also necessary to mention whether there was any difference in response to LPS between twins and singletons, as has been the case in previous reports.” 

Only 3/8 fetus were singletons, 5/8 were twins in LPS group. Statistical analyses are not possible with only 3 fetuses.

Below, in this answer, we added a figure with the evolution of heart rate variability measures in LPS group for each fetus. We added complete data about each fetus in supporting information. Changes in singletons and twins seem similar.

We added in limits (l409): =>Third, due to small number of fetal sheep, it was not possible to research eventual differences in HRV modifications between singletons and twins.

Is the p-value of H2 for FSI a misstatement? Yes, it was a misstatement

Correction: table 2: (FSI) p at H2: 100 -> 1.00

Reviewer 2

“Line66-68: Authors describe “Unfortunately, current understanding of how infection/inflammation affects fetal HRV is limited, in part due to the heterogeneous nature of clinical infections [15].

While I, as an obstetrician and gynecologist, can very much understand what is being conveyed, I think it would be better to rephrase this statement to reflect that the diagnostic criteria for intrauterine infection or inflammation are not unified across pathology, biochemistry, and clinical judgment. 

Also, the transition from this statement to lines 69-70 is abrupt. It should be described why a hypothesis was formulated that HRV analysis would be useful for early detection.”

For better readability, we change these sentences: (l63-66)

We removed : “Unfortunately, current understanding of how infection/inflammation affects fetal HRV is limited, in part due to the heterogeneous nature of clinical infections [15].” For better readability, we finally chose to not discuss the diversity of fetal infection and the variability of definition. In fact, our model focus on fetal inflammatory response syndrome, which is clearly defined and not on intrauterine infection and inflammation. 

We added =>Since HRV is correlated with SNA alterations and since it allows detection of neonatal inflammation, we hypothesized that HRV analysis can be used for early detection of acute FIRS”.

In Response to reviewers files :

Evolution of heart rate variability measures in LPS group for each fetus

Blue Singleton (n=3); Green Twin (n=5) at baseline (H0), H1, H2, H3, H4, H5, H6 after LPS injection

SDNN = standard deviation of normal to normal, DFA = detrended fluctuation analysis, SD2 = Standard Deviation 2, LTV = long term variability, ApEn = Approximate Entropy

---

## [Decision Letter · Decision Letter 1]

25 Aug 2023

PONE-D-23-13482R1Early heart rate variability changes during acute fetal inflammatory response syndrome: an experimental study in a fetal sheep modelPLOS ONE

Dear Dr. Chevalier, 

Thank you for submitting response to reviewers comment.. After careful consideration, we feel that it has merit but does not fully meet PLOS ONE’s publication criteria as it currently stands. Therefore, we invite you to submit a revised version of the manuscript that addresses the points raised during the review process.

Please submit your revised manuscript by Oct 09 2023 11:59PM.  If you will need more time than this to complete your revisions, please reply to this message or contact the journal office at plosone@plos.org. Please include the following items when submitting your revised manuscript:A rebuttal letter that responds to each point raised by the academic editor and reviewer(s). You should upload this letter as a separate file labeled 'Response to Reviewers'.A marked-up copy of your manuscript that highlights changes made to the original version. You should upload this as a separate file labeled 'Revised Manuscript with Track Changes'.An unmarked version of your revised paper without tracked changes. You should upload this as a separate file labeled 'Manuscript'.

We look forward to receiving your revised manuscript.

Kind regards,

Sanjoy Kumer Dey, M.D

Academic Editor

PLOS ONE

Reviewers' comments:

Reviewer's Responses to Questions

**Comments to the Author**

1. If the authors have adequately addressed your comments raised in a previous round of review and you feel that this manuscript is now acceptable for publication, you may indicate that here to bypass the “Comments to the Author” section, enter your conflict of interest statement in the “Confidential to Editor” section, and submit your "Accept" recommendation.

Reviewer #1: (No Response)

Reviewer #2: All comments have been addressed

2. Is the manuscript technically sound, and do the data support the conclusions?

Reviewer #1: Partly

Reviewer #2: Yes

3. Has the statistical analysis been performed appropriately and rigorously? 

Reviewer #1: Yes

Reviewer #2: Yes

4. Have the authors made all data underlying the findings in their manuscript fully available?

Reviewer #1: No

Reviewer #2: Yes

5. Is the manuscript presented in an intelligible fashion and written in standard English?

Reviewer #1: Yes

Reviewer #2: Yes

6. Review Comments to the Author

Reviewer #1: Your modifications to the changes you have made are greatly welcome. There are still many unknowns in the area of fetal physiology, and hence the interpretation of study results needs to be more cautious. In particular, the parameters of fetal heart rate variability are often confused with the interpretation of study results for adults, which requires caution.

This revision requires further review of the discussion part of your paper.

The first half of the section is a duplication of what is described in the results section. Rather than duplicate the results section, you should provide a discussion of the FHR and the large discrepancy in the timing of activation of various parameters such as SDNN. If the response is due to sympathetic nerves, I have the impression that the FHR should increase with the activation of SDNN.

In addition, it should be further emphasized that this model is a "non-shockable inflammation exposure model. Describing such a model as a "sepsis model" may be misleading to the reader.

The absence of effects of cholinergic anti-inflammatory pathways in this study may be due to the fact that such inflammation was mild. The discussion section is poorly written in this revision. A comparison with a study of acute exacerbated inflammation (PMID 35110628) would add depth to the discussion.

In summary, we request that the results of this peer review (1) discussion of inconsistencies in the timing of activation of various parameters (2) comparison with changes in parameters in severe inflammatory exposure to the fetus.

Reviewer #2: Thank you for giving me the opportunity for re-evaluation. The paper has been well revised.　I consider this scientific paper to be sufficient for publication.

7. PLOS authors have the option to publish the peer review history of their article (what does this mean?). If published, this will include your full peer review and any attached files.

Reviewer #1: No

Reviewer #2: No

---

## [Author Response · Author response to Decision Letter 1]

31 Aug 2023

We would like to thank you for giving us the opportunity to submit a corrected version of our work in your journal.

We also thank the reviewers for the time spent judging our manuscript and the quality of their comments. We have responded to each of the comments and believe that the quality of the manuscript is improved by the suggestions made to us.

“you should provide a discussion of the FHR and the large discrepancy in the timing of activation of various parameters such as SDNN. If the response is due to sympathetic nerves, I have the impression that the FHR should increase with the activation of SDNN.” “we request that the results of this peer review (1) discussion of inconsistencies in the timing of activation of various parameters”.

It is indeed unexpected that HRV changes normalize at the same time as tachycardia appears. It has already been shown that initial increase in HRV can be followed by suppression of time-domain measures [PMID 35110628, 24944248, 18456675]. However, we agreed that with the onset of tachycardia, we might have expected an increase of indices usually associated with sympathetic activity including SDNN. We have no pathophysiological explanation for this discrepancy.

We have added this following paragraph (l363-368):

“It is also unexpected that the onset of tachycardia from H5 was associated with a return to baseline of HRV markers. It has already been shown that initial increase in HRV can be followed by suppression of time-domain measures [27, 28, 29, 31]. However, with the onset of tachycardia, we might have expected an increase of indices usually associated with sympathetic activity. We have no pathophysiological explanation for this discrepancy."

“it should be further emphasized that this model is a "non-shockable inflammation exposure model. Describing such a model as a "sepsis model" may be misleading to the reader.”

We agree that our model is not a sepsis model. In contrast to Durosier et al (PMID: 26290042), who used exactly the same model, we have chosen to not describe it as “sepsis”. We describe the model as a “fetal inflammatory response syndrome” as described by Gomez et al (PMID 9704787, 17762416) by an elevation of fetal interleukin 6. We described it in introduction (l39) and in discussion (l315).” In material and methods (l104) and in discussion (l318) we have specified that our FIRS model was without shock or cardiovascular decompensation.

To avoid misleading we add in abstract (l29-30): “In our FIRS model without shock or cardiovascular decompensation”

“The absence of effects of cholinergic anti-inflammatory pathways in this study may be due to the fact that such inflammation was mild. The discussion section is poorly written in this revision. A comparison with a study of acute exacerbated inflammation (PMID 35110628) would add depth to the discussion.” “we request that the results of this peer review (2) comparison with changes in parameters in severe inflammatory exposure to the fetus”

We agree that absence of effects of cholinergic anti-inflammatory pathways may be due to the fact that such inflammation was mild. 

We have added this paragraph (l353-359): We can hypothesize that cholinergic anti-inflammatory pathway was not activated because of insufficiently strong inflammation. Indeed, Magawa et al induced acute on chronic inflammation on preterm fetal sheep with LPS bolus injections starting 48 h after LPS IV infusion. After a first LPS bolus, there was stronger hemodynamic response to inflammation with hypotension and tachycardia associated with initial increase in most FHRV measures, including RMSSD and HF, consistent with an activation of cholinergic anti-inflammatory pathway [PMID 35110628].

We have also added this sentence (l406): This experiment was not without limitations. First, it focused specifically on acute onset FIRS in the near-term fetus and the model does not therefore reflect the entire range of fetal infections, which may be subacute or chronic with differing intensity.

---

## [Decision Letter · Decision Letter 2]

18 Sep 2023

PONE-D-23-13482R2Early heart rate variability changes during acute fetal inflammatory response syndrome: an experimental study in a fetal sheep modelPLOS ONE

Dear Dr. Geoffroy Chevalier,

Thank you for submitting your manuscript to PLOS ONE. After careful consideration, we feel that it has merit but does not fully meet PLOS ONE’s publication criteria as it currently stands. Therefore, we invite you to submit a revised version of the manuscript that addresses the points raised during the review process.

We look forward to receiving your revised manuscript.

Kind regards,

Sanjoy Kumer Dey, M.D

Academic Editor

PLOS ONE

Reviewers' comments:

Reviewer's Responses to Questions

**Comments to the Author**

1. If the authors have adequately addressed your comments raised in a previous round of review and you feel that this manuscript is now acceptable for publication, you may indicate that here to bypass the “Comments to the Author” section, enter your conflict of interest statement in the “Confidential to Editor” section, and submit your "Accept" recommendation.

Reviewer #1: All comments have been addressed

2. Is the manuscript technically sound, and do the data support the conclusions?

Reviewer #1: Partly

3. Has the statistical analysis been performed appropriately and rigorously? 

Reviewer #1: I Don't Know

4. Have the authors made all data underlying the findings in their manuscript fully available?

Reviewer #1: Yes

5. Is the manuscript presented in an intelligible fashion and written in standard English?

Reviewer #1: Yes

6. Review Comments to the Author

Reviewer #1: I agree with the changes you have made. The change in the discussion section is also a good one.

However, the timing discrepancy between the occurrence of tachycardia and changes in heart rate variability parameters requires further consideration.

Please review the response time lag between these blood pressure and heart rate changes and HRV parameters in the past literature and recognize how much this study deviates from the previous study. Possible causes should then be described. Because if this is due to noise generated at a particular time, it would undermine the interpretation of the study results.

Also, please pay up-to-date attention to typos and misstatements and review the manuscript again.

Ex.L260 LTV = short-term variation

7. PLOS authors have the option to publish the peer review history of their article (what does this mean?). If published, this will include your full peer review and any attached files.

Reviewer #1: No

---

## [Author Response · Author response to Decision Letter 2]

5 Oct 2023

We thank the reviewers for the time spent judging our manuscript and the quality of their comments. 

We have responded to the two comments.

Reviewer’s comments:

“The timing discrepancy between the occurrence of tachycardia and changes in heart rate variability parameters requires further consideration.

Please review the response time lag between these blood pressure and heart rate changes and HRV parameters in the past literature and recognize how much this study deviates from the previous study. Possible causes should then be described. Because if this is due to noise generated at a particular time, it would undermine the interpretation of the study results.”

We agree that the timing discrepancy between the occurrence of tachycardia and changes in HRV parameters requires further discussion:

Concerning the significance of heart rate and blood pressure changes: 

We used the same model of FIRS as Durosier et al (1). In their study, fetal heat rate (FHR) was significantly higher in LPS group at H6 in comparison to control group. Whereas, no significative change was found at H0, H1 and H3. We observed similar hemodynamic changes: FHR was significantly higher at H5 (196 bpm [182; 209], p = 0.01) and H6 (203 bpm [187; 223], p = 0.01) compared with baseline (173 bpm [170; 178]). In comparison to control group, FHR was significantly higher at H6. 

Concerning mean arterial pressure (MAP), our results differ slightly from those of Durosier et al. In Durosier et al study, no significative change was found in MAP in comparison to control group. In our study, MAP was significantly lower only at H6 in LPS group (41.0 mmHg [40.0; 42.2]) compared with control group (47.0 mmHg [43.0; 55.5], p = 0.04). However MAP was not significantly decrease at H6 (41.0 mmHg [40.0; 42.2], p = 0.07) compared with baseline (45.5 mmHg [43.7; 48.5]).

Concerning discrepancy between the occurrence of tachycardia and changes in heart rate variability parameters: 

It has been shown that HRV increase can be concomitant with tachycardia (2–4). In these studies, fetuses were premature, and fetal reactions were stronger with strong hypotension.

However, discrepancy between the occurrence of tachycardia and changes in HRV parameters has already been reported in fetal sheep at term after LPS injection (1,5). Kyozuka has also reported changes in HRV parameters in fetal sheep near term with no tachycardia (5). In their model LPS was infused into the amniotic cavity for 2 days following the 4th postoperative day to develop histological chorioamniotis. There were no significant differences in FHR after LPS injection until intra uterine fetal death. Only one HRV parameter was studied (STV: Short term variability quantifying the difference between 2 consecutive beats). This HRV parameter was significantly increase at 6, 4, and 3 h before intra uterine fetal death with no tachycardia.

Durosier el al measured 44 fHRV continuously every 5 min using continuous individualized multi-organ variability analysis (CIMVA) (1). CIMVA created a fetal HRV measures matrix across five signal-analytical domains, thus describing complementary properties of fetal HRV. Using principal component analysis (PCA), they derived and quantitatively compared the CIMVA fetal heart rate variability PCA signatures of inflammatory response in LPS and control groups.

Qualitatively, the discrimination between LPS-injected animals and control animals occurs between 2 h and 3 h post injection. Whereas FHR was significantly higher in LPS group only at H6, in comparison to control group. 

It should be noted that the difference in HRV between the 2 groups persisted with the onset of tachycardia, which is not the case in our study. This may be explained by the greater sensitivity of the combined use of HRV markers compared with markers evaluated separately. Nevertheless, our results offer hope that use of HRV markers could allow detection of FIRS before the onset of tachycardia. Our results are therefore consistent with those in the literature.

We have highlighted an overall increase in global variability. As discussed in previous reviewing, inhibition or activation of autonomic activities cannot be clearly established by our HRV indices changes. We have no clear pathophysiological explanation for the discrepancy between the occurrence of tachycardia and changes in heart rate variability parameters.

We add: (l338-348)

“It has been shown that HRV increase can be concomitant with tachycardia (2–4). In these studies, fetuses were premature, and fetal reactions were stronger with strong hypotension. However, discrepancy between the occurrence of tachycardia and changes in HRV parameters has already been reported in two studies in fetal sheep at term after LPS injection (1,5). First, Durosier et al found that discrimination between LPS-injected animals and control group using HRV occurs between 2 h and 3 h post injection. Whereas FHR was significantly higher in LPS group only at H6. Second, Kyozuka et al., who studied short-term variability changes after LPS infusion into the amniotic cavity, found significative change in short-term variability, with no tachycardia, at 6, 4, and 3 h before intrauterine fetal death (5).”

“please pay up-to-date attention to typos and misstatements”

we have proofread the entire manuscript.

We change: 

LTV = short -> long-term variability, (l260, l546 and in S. table 2)

1. Durosier LD, Herry CL, Cortes M, Cao M, Burns P, Desrochers A, et al. Does heart rate variability reflect the systemic inflammatory response in a fetal sheep model of lipopolysaccharide-induced sepsis? Physiol Meas. oct 2015;36(10):2089‑102. 

2. Lear CA, Davidson JO, Booth LC, Wassink G, Galinsky R, Drury PP, et al. Biphasic changes in fetal heart rate variability in preterm fetal sheep developing hypotension after acute on chronic lipopolysaccharide exposure. Am J Physiol Regul Integr Comp Physiol. 15 août 2014;307(4):R387-395. 

3. Magawa S, Lear CA, Beacom MJ, King VJ, Kasai M, Galinsky R, et al. Fetal heart rate variability is a biomarker of rapid but not progressive exacerbation of inflammation in preterm fetal sheep. Sci Rep. 2 févr 2022;12:1771. 

4. Blad S, Welin AK, Kjellmer I, Rosén KG, Mallard C. ECG and heart rate variability changes in preterm and near-term fetal lamb following LPS exposure. Reprod Sci Thousand Oaks Calif. juill 2008;15(6):572‑83. 

5. Kyozuka H, Yasuda S, Hiraiwa T, Nomura Y, Fujimori K. The change of fetal heart rate short-term variability during the course of histological chorioamnionitis in fetal sheep. Eur J Obstet Gynecol Reprod Biol. sept 2018;228:32‑7.

---

## [Editor Report · Decision Letter 3]

23 Oct 2023

Early heart rate variability changes during acute fetal inflammatory response syndrome: an experimental study in a fetal sheep model

PONE-D-23-13482R3

Dear Dr.Geoffroy Chevalier ,

We’re pleased to inform you that your manuscript has been judged scientifically suitable for publication and will be formally accepted for publication once it meets all outstanding technical requirements.

Kind regards,

Sanjoy Kumer Dey, M.D

Academic Editor

PLOS ONE

---

## [Editor Report · Acceptance letter]

17 Nov 2023

PONE-D-23-13482R3 

Early heart rate variability changes during acute fetal inflammatory response syndrome: an experimental study in a fetal sheep model 

Dear Dr. Chevalier:

I'm pleased to inform you that your manuscript has been deemed suitable for publication in PLOS ONE. Congratulations! Your manuscript is now with our production department. 

Kind regards, 

on behalf of

Dr. Sanjoy Kumer Dey 

Academic Editor

PLOS ONE